# Differential Privacy on Fully Dynamic Streams

**Yuan Qiu**
School of Cyber Science and Engineering
Southeast University
Nanjing, China 211189
yuanqiu@seu.edu.cn

**Ke Yi**
Department of Computer Science and Engineering
Hong Kong University of Science and Technology
Hong Kong SAR, China 999077
yike@cse.ust.hk

## Abstract

A fundamental problem in differential privacy is to release privatized answers to a class of linear queries with small error. This problem has been well studied in the static case. In this paper, we consider the fully dynamic setting where items may be inserted into or deleted from the dataset over time, and we need to continually release query answers at every time instance. We present efficient black-box constructions of such dynamic differentially private mechanisms from static ones with only a polylogarithmic degradation in the utility.

## 1 Introduction

In the data streaming model, a stream consists of a pair of possibly infinitely long vectors $(\vec{s}, \vec{x})$. It defines a dynamically changing dataset $D_t$, which is a multiset of elements from a universe $\mathcal{X}$. Initially, $D_0 = \emptyset$. At time $t \in \mathbb{N}^+$, $(s_t, x_t)$ arrives, where $s_t \in \{+, -, \bot\}$ indicates the operation and $x_t \in \mathcal{X}$ is the relevant element. Then $D_t$ is defined inductively as follows:

- If $s_t = +$, then $D_t := D_{t-1} \uplus \{x_t\}$, i.e., one copy of $x_t$ is inserted into the dataset.
- If $s_t = -$, then $D_t := D_{t-1} - \{x_t\}$, i.e., one copy of $x_t$ is deleted from the dataset (it is required that at least one copy of $x_t$ exists in $D_{t-1}$).
- If $s_t = \bot$, then $D_t := D_{t-1}$.

Two variants of the streaming model have been considered in the literature [25, 5]: The model above is referred to as the fully dynamic setting (a.k.a. the *turnstile model*), while the case where $s_t$ can only be $+$ or $\bot$ is called the insertion-only setting (a.k.a. the *cash register model*).

A fundamental problem in differential privacy (DP) is how to answer a class of *linear queries* on a given dataset $D$ privately and accurately. In particular, the *private multiplicative weights mechanism (PMW)* [15, 17] can answer a set of arbitrary linear queries with maximum error[1] $\tilde{O}(\sqrt{|D|})$. Better error bounds can be achieved if the class of queries exhibit good structures. For instance, $d$-dimensional half-space queries can be answered with error $\tilde{O}(|D|^{\frac{1}{2} - \frac{1}{2d}})$ [26]; please see Appendix A for a review of results on various query classes. Note that, since there can be many queries in the class, these algorithms do not return the query answers explicitly; instead, a privatized data structure (e.g., a histogram in the case of PMW) is often returned, and a data analyst can subsequently extract the answer of any query from the data structure.

Combining the query-answering problem and the data streaming model naturally gives rise to the problem of *differential privacy under continual observation*, first studied in the foundational work of Dwork et al. [9]. Here, the goal is to continually release privatized query answers under the following requirements:

---

[1]The $\tilde{O}$ notation suppresses dependencies on the privacy parameter $\varepsilon$ and polylogarithmic terms.

39th Conference on Neural Information Processing Systems (NeurIPS 2025).

- Online: For any $t$, we must release a data structure $\mathcal{M}^{(t)}(D_t)$ before $(s_{t+1}, x_{t+1})$ arrives.

- Private: All released data structures $(\mathcal{M}^{(1)}(D_1), \mathcal{M}^{(2)}(D_2), \dots)$ jointly satisfy DP.

- Accurate: For every $t$, all queries on $D_t$ can be answered from $(\mathcal{M}^{(1)}(D_1), \dots, \mathcal{M}^{(t)}(D_t))$ with small error, ideally matching the error in the static case, i.e., as if all queries were answered on $D_t$ one-shot.

- Efficient: The mechanism should run efficiently.

The pioneer work [9, 4] has only studied the basic counting problem, i.e., the query just asks for $|D_t|$, under the insertion-only streaming model. However, we observe that their algorithms are actually black-boxed, and can be instantiated with any static mechanism for some other query class, as long as the queries are union-preserving, with a polylogarithmic-factor increase in the error bound. Formally, a query $f$ is called union-preserving if $f(D^{(1)} \uplus D^{(2)}) = f(D^{(1)}) + f(D^{(2)})$ for any datasets $D^{(1)}, D^{(2)}$, where $\uplus$ denotes the multiset union. For example, by plugging in PMW, any class of linear queries on $D_t$ can be answered with error $\tilde{O}(\sqrt{|D_t|})$ for every $t$. Perhaps not realizing this, [6] presented a white-box version of PMW for the insertion-only streaming model, but the error is $\tilde{O}(|D_t|^{3/4})$. They also showed a black-box solution, but the error is $\tilde{O}(|D_t|^{5/6})$ when instantiated with PMW.

A standard approach to dealing with the fully-dynamic case is to divide the stream into two insertion-only steams: one only containing insertions and one only containing deletions but treating these deletions as insertions. Let $D_t^+ := \uplus_{i \le t : s_i = +} \{x_i\}$ be all items inserted up to time $t$, and $D_t^- := \uplus_{i \le t : s_i = -} \{x_i\}$ all items deleted up to time $t$, then $D_t = D_t^+ - D_t^-$. For any union-preserving query $f$, we have $f(D_t) = f(D_t^+) - f(D_t^-)$, so we can run two separate instances of the insertion-only mechanism, and use their difference to answer $f$ on $D_t$. However, the worst-case error of this solution is very bad. Suppose we instantiate the insertion-only algorithm of [9, 4] with PMW. Then $f(D_t^+)$ and $f(D_t^-)$ can be answered with error $\tilde{O}\left(\sqrt{|D_t^+|}\right)$ and $\tilde{O}\left(\sqrt{|D_t^-|}\right)$, respectively, which means that the error for $f(D_t)$ is $\tilde{O}\left(\sqrt{|D_t^+| + |D_t^-|}\right)$. This can be arbitrarily worse than the optimal error of $\tilde{O}(\sqrt{|D_t|})$ in the static case: $D_t^+$ and $D_t^-$ may both be very large but $D_t = \emptyset$.

**Our Contributions**. In this paper, we present a black-box algorithm for fully-dynamic streams that can answer any class of union-preserving queries with error matching that in the static case, up to polylogarithmic factors. For instance, when instantiated with PMW, our algorithm achieves an error of $\tilde{O}(\sqrt{|D_t|})$ for every $t$. Furthermore, its total running time up to time $t$ is also just a polylogarithmic factor higher than that of the static mechanism run on all the stream elements up to time $t$.

## 2  Preliminaries

### 2.1  Differential Privacy

Let $\mathcal{X}$ be the universe of items. A static dataset is a multiset of items $D \in \mathbb{N}^{\mathcal{X}}$. Two static datasets $D, D' \in \mathbb{N}^{\mathcal{X}}$ are *neighbors*, denoted $D \sim D'$, if there exists an item $x \in \mathcal{X}$, such that $D = D' \uplus \{x\}$ or vise versa. *Differential privacy (DP)* [8] is defined as follows.

**Definition 1** (Differential Privacy [8])**.** *A randomized mechanism $\mathcal{M} : \mathbb{N}^{\mathcal{X}} \to \mathcal{Y}$ satisfies $(\varepsilon, \delta)$-DP if for any neighboring datasets $D \sim D'$ and any subset of outputs $Y \subseteq \mathcal{Y}$,*

$$\Pr[\mathcal{M}(D) \in Y] \le e^{\varepsilon} \cdot \Pr[\mathcal{M}(D') \in Y] + \delta. \tag{1}$$

In the streaming setting, two streams are neighbors if one has one more update, which can be either an insertion or a deletion, than the other [9]. Formally, two streams $(\vec{s}, \vec{x})$ and $(\vec{s}', \vec{x}')$ are neighbors, if $\exists i \in \mathbb{N}^+$ such that $(s_j, x_j) = (s_j', x_j')$ for any $j \ne i$, and either $s_i$ or $s_i'$ is $\perp$. Plugging this neighboring relationship into Definition 1 yields the DP definition for the streaming model. More precisely, $\mathcal{M}(D)$ and $\mathcal{M}(D')$ in (1) are replaced by $(\mathcal{M}^{(1)}(D_1), \mathcal{M}^{(2)}(D_2), \dots)$ and $(\mathcal{M}^{(1)}(D_1'), \mathcal{M}^{(2)}(D_2'), \dots)$ respectively, where $D_i$ and $D_i'$ are the datasets induced by any two neighboring streams at time $i$.

An important property used in designing DP mechanisms is composition, which comes in two settings:

**Theorem 2** (Sequential Composition [8]). *Let $\mathcal{M}_i : \mathbb{N}^{\mathcal{U}} \to \mathcal{Y}_i$ each be an $(\varepsilon_i, \delta_i)$-DP mechanism. Then the composed mechanism $\mathcal{M}(D) = (\mathcal{M}_1(D), \ldots, \mathcal{M}_k(D))$ is $(\sum_{i=1}^{k} \varepsilon_i, \sum_{i=1}^{k} \delta_i)$-DP.*

Note that there are many "advanced" versions of sequential composition [10, 21, 2] with better dependencies on $k$. Nevertheless, as $k$ is logarithmic in all of our algorithms, those advanced versions do not offer better bounds for the problem studied in this paper.

**Theorem 3** (Parallel Composition [24]). *Let $\mathcal{U} = \mathcal{U}_1 \cup \cdots \cup \mathcal{U}_k$ be a partitioning of the universe[2] $\mathcal{U}$, and $\mathcal{M}_i : \mathbb{N}^{\mathcal{U}_i} \to \mathcal{Y}_i$ each be an $(\varepsilon_i, \delta_i)$-DP mechanism. Then the composed mechanism $\mathcal{M}(D) = (\mathcal{M}_1(D \cap \mathcal{U}_1), \ldots, \mathcal{M}_k(D \cap \mathcal{U}_k))$ is $(\max_{i=1}^{k} \varepsilon_i, \max_{i=1}^{k} \delta_i)$-DP.*

### 2.2 Linear Queries

A *linear query* is specified by a function $f : \mathcal{X} \to [0, 1]$. The result of evaluating $f$ on $D$ is defined as $f(D) := \sum_{x \in D} f(x)$. A fundamental problem in differential privacy is the following: Given a set of linear queries $\mathcal{F} = \{f_1, \ldots, f_{|\mathcal{F}|}\}$, design a DP mechanism $\mathcal{M}$ that, on any given $D$, outputs a data structure $\mathcal{M}(D)$, from which an approximate $\hat{f}_{\mathcal{M}}(D)$ can be extracted for any $f \in \mathcal{F}$. We say that $\mathcal{M}$ has error $\alpha$ with probability $1 - \beta$, if

$$\Pr\left[\max_{f \in \mathcal{F}} \left|\hat{f}_{\mathcal{M}}(D) - f(D)\right| > \alpha\right] \leq \beta$$

for any $D$, where the probability is taken over the internal randomness of $\mathcal{M}$. Clearly, $\alpha$ is a function of the privacy parameters $(\varepsilon, \delta)$ and the failure probability $\beta$. For most mechanisms, it also depends on the data size $|D|$, number of queries $|\mathcal{F}|$, and domain size $|\mathcal{X}|$. To simplify notation, we often omit some of these parameters from the full list $\alpha(\varepsilon, \delta, \beta, |D|, |\mathcal{F}|, |\mathcal{X}|)$ if they are clear from the context. Similarly, we denote the running time of the mechanism by $\mathrm{time}(|D|, \cdot)$, which depends on the data size $|D|$ and possibly other parameters.

There is extensive work [30, 15, 17, 29, 14, 1, 22, 23, 26] on the best achievable $\alpha$ for various families of linear queries. This paper takes a black-box approach, i.e., we present dynamic algorithms that can work with any $\mathcal{M}$ that has been designed for queries $\mathcal{F}$ on a static dataset $D$. The error for the dynamic algorithm will be stated in terms of the $\alpha$ function of the mechanism $\mathcal{M}$ that is plugged into the black box. Nevertheless, we often derive the explicit bounds for the following two most interesting and extreme cases:

**Basic counting.** If $\mathcal{F}$ consists of a single query $f(\cdot) \equiv 1$, which simply returns $f(D) = |D|$, then the "data structure" $\mathcal{M}(D)$ consists of just one number, which is a noise-masked $f(D)$. The most popular choice of the noise is a random variable drawn from the Laplace distribution $\mathrm{Lap}(\frac{1}{\varepsilon})$. Its error function $\alpha$ is given by $\alpha_{\mathrm{Lap}}(\varepsilon, \beta) = O\left(\frac{1}{\varepsilon} \log \frac{1}{\beta}\right)$. Alternatively, one can add a Gaussian noise, which is $(\varepsilon, \delta)$-DP for $\delta > 0$ and yields $\alpha_{\mathrm{Gauss}}(\varepsilon, \delta, \beta) = O\left(\frac{1}{\varepsilon} \sqrt{\log \frac{1}{\delta} \log \frac{1}{\beta}}\right)$. The two error bounds are generally incomparable, but the former is usually better since $\delta \leq \beta$ in common parameter regimes. Both the Laplace and the Gaussian mechanism have $\mathrm{time}(|D|) = O(|D|)$.

**Arbitrary queries.** If $\mathcal{F}$ is a set of arbitrary linear queries, then the *private multiplicative weights (PMW)* [15, 17] mechanism achieves $\alpha_{\mathrm{PMW}} = \tilde{O}(\sqrt{|D|})$ for $\delta > 0$ and $\tilde{O}(|D|^{2/3})$ for $\delta = 0$. These error bounds are optimal for $|\mathcal{F}|$ sufficiently large. The running time of PMW is $\tilde{O}(|D| + |\mathcal{X}| \cdot |\mathcal{F}| \cdot |D|^2/\alpha^2)$.

There are many possibilities between the two extreme cases, and the achievable error bound $\alpha$ intricately depends on the discrepancy of the query set $\mathcal{F}$ [16, 26]. We include a brief review in Appendix A, which is not necessary for the understanding of this paper. We make a reasonable assumption that $\alpha$ does not depend on any of its parameters exponentially, which allows us to ignore the constant coefficients in the parameters, e.g., $\alpha(O(\varepsilon))$ is asymptotically the same as $\alpha(\varepsilon)$. This assumption holds for most existing mechanisms for linear queries.

---

[2]Note that the "universe" $\mathcal{U}$ above is not necessarily the same as the universe $\mathcal{X}$ of stream items. In fact, in our algorithms, we often take $\mathcal{U}$ as the time domain, and parallel composition then implies that running an $(\varepsilon, \delta)$-DP mechanism in each disjoint time interval satisfies $(\varepsilon, \delta)$-DP over the entire stream.

## 2.3  Differential Privacy for Insertion-only Streams

The insertion-only algorithms, as well as our fully dynamic algorithm, work by decomposing $D = D^{(1)} \uplus \cdots \uplus D^{(k)}$ for some $k$, so $f(D) = f(D^{(1)}) + \cdots + f(D^{(k)})$ by the union-preserving property. Thus, if the error of each $f(D^{(i)})$ is $\alpha(\cdot, \beta)$, the error of $f(D)$ is at most $k \cdot \alpha(\cdot, \beta/k)$ by a union bound. Further, if the estimates of $f(D^{(i)})$ are unbiased with good concentration properties, this bound can be tightened. For example, for the basic counting query $f(D) = |D|$, if we use the Laplace mechanism to estimate each $f(D^{(i)})$, then the error for $f(D)$ is $O\left(\frac{1}{\varepsilon}\left(\sqrt{k \log \frac{1}{\beta}} + \log \frac{1}{\beta}\right)\right)$, which is better than $k \cdot \alpha_{\text{Lap}}(\cdot, \beta/k) = O\left(\frac{k}{\varepsilon} \log \frac{k}{\beta}\right)$. We derive this result, as well as the error bounds of some other mechanisms under such a disjoint union, in Appendix B. In the main text, for generality we will use $\alpha^{(k)}$ to denote this error bound. Note that $\alpha^{(k)}$ is also a function of $|D|, \varepsilon, \delta, \beta, |\mathcal{F}|, |\mathcal{X}|$, but we may omit these parameters when the context is clear. Using this notation, the result of existing insertion-only algorithms can be restated as follows.

**Lemma 4** ([4]). *For a set of union-preserving queries $\mathcal{F}$, let $\mathcal{M} : \mathbb{N}^{\mathcal{X}} \to \mathcal{Y}$ be an $(\varepsilon, \delta)$-DP static mechanism whose error is $\alpha(\varepsilon, \delta, \beta, |D|, |\mathcal{F}|, |\mathcal{X}|)$ on any $D$, with running time $\text{time}(|D|)$. Then there is an $(\varepsilon, \delta)$-DP mechanism $\mathcal{M}_{\mathcal{F}, ins}$ for insertion-only streams, so that at any time $t$, it answers queries $\mathcal{F}$ on $D_t$ with error*

$$\gamma(t; \varepsilon, \delta, \beta, |D_t|, |\mathcal{F}|, |\mathcal{X}|) = \alpha^{(\log t)}\left(\frac{\varepsilon}{\log t}, \frac{\delta}{\log t}, \beta, |D_t|, |\mathcal{F}|, |\mathcal{X}|\right) .$$

*The total running time[3] of $\mathcal{M}_{\mathcal{F}, ins}$ up to time $t$ is $O(\log t) \cdot \text{time}(|D_t|)$.*

Our algorithm will also use a special case of Lemma 4 when $\mathcal{F}$ is the basic counting query. Plugging in our bound on $\alpha_{\text{Lap}}^{(k)}$ in Appendix B leads to the following explicit error bound, which is slightly better than that in [4].

**Corollary 1.** *There is an $\varepsilon$-DP mechanism $\mathcal{M}_{cnt, ins}$ that at any time $t$, answers the counting query $f(D_t) = |D_t|$ on an insertion-only stream with error*

$$\gamma_{cnt}(t; \varepsilon, \beta) = \alpha_{\text{Lap}}^{(\log t)}\left(\frac{\varepsilon}{\log t}, \beta\right) = O\left(\frac{\log^{1.5} t}{\varepsilon}\sqrt{\log \frac{1}{\beta}} + \frac{\log t}{\varepsilon}\log \frac{1}{\beta}\right) .$$

# 3  Differential Privacy on Fully Dynamic Streams

In this section, we describe our algorithm for answering a set of union-preserving queries on a possibly infinite fully dynamic stream under differential privacy. We use $n_t := |D_t|$ to denote the size of the dataset at time $t$, and $N_t$ to denote the total number of updates up to time $t$. Note that for an insertion-only stream we have $n_t = N_t$, but on a fully dynamic stream $n_t$ may be much smaller than $N_t$. Our goal is to achieve an error bound that depends on $n_t$.

We will treat a fully dynamic stream as a set of labeled time intervals. An interval $[i, j)$ labeled with item $x \in \mathcal{X}$ represents that a copy of $x$ is inserted at time $i$ and deleted at time $j$. It is possible that $j = \infty$, if the item is never deleted. Note that this interval representation of a stream is not unique, e.g., when many copies of the same item are inserted and then deleted. Any representation can be used; in fact, our algorithm does not use this interval representation, only the analysis does.

Using the interval representation, $D_t$ consists of all items whose intervals are stabbed by $t$, i.e., all intervals $[i, j) \ni t$. We will make use of the *interval tree* [7] to organize all the intervals. In an interval tree, each $D_t$ is decomposed into a logarithmic number of subsets, each of which consists of one-sided intervals, which will allow us to use the insertion-only mechanism. However, there are two technical difficulties in implementing such a plan. First, the intervals are given in an online fashion, i.e., at time $t$, we only see the endpoints of the intervals prior to $t$. When we see the left endpoint of an interval, we do not know where in the interval tree to put this interval, yet, we need to immediately release privatized information about this interval. Second, the interval tree on an infinite stream is also infinitely large, so we have to build it incrementally, while allocating the privacy budget appropriately.

---

[3]For running time, it is often assumed that no-ops in the stream can be skipped at no cost.

We describe how to overcome the difficulties above in Section 3.1. In Section 3.2, we introduce a DP mechanism running at each node of the new tree structure to support querying at any time with respect to intervals stored in the tree. The output of the whole mechanism is obtained by combining the individual mechanisms at the tree nodes, which is summarized in Section 3.3.

## 3.1 Online Interval Tree

We first build a binary tree $\mathcal{T}$ over the timestamps, where each timestamp corresponds to a tree node. Figure 1 shows an (offline) interval tree built on the first 8 timestamps. It is clear that the tree on timestamps $\{1, \ldots, t\}$ has $t$ nodes and $O(\log t)$ height. In the online setting, $\mathcal{T}$ will grow from left to right. We order the nodes using an in-order traversal of $\mathcal{T}$: $v_1, v_2, \ldots$, and we will release the information about $v_i$ right after time $i$.

Ignoring differential privacy for now, let us first focus on how to answer a stabbing query using an interval tree, i.e., report all items (intervals) in the dataset $D_q$ at query time $q$. In a standard interval tree, an interval is stored at the *highest* node $v_t$ such that its timestamp $t$ stabs the interval. Denote by $D(v)$ the set of labeled intervals stored at $v$. For example, $D(v_4) = \{a, b, c\}$ in Figure 1. Given a query at time $q$, we follow the root-to-node path to $v_q$ in $\mathcal{T}$. For each *left* node $v_l$ on the path where $l \leq q$, we find all intervals $[i, j)$ in $D(v_l)$ such that $j > q$; for each *right* node $v_r$ on the path where $r > q$, we find all intervals $[i, j)$ in $D(v_r)$ such that $i \leq q$. Standard analysis on the interval tree shows that these subsets form a disjoint union of all intervals stabbed by $q$.

**Example 1** (Interval Tree Query). *Given the interval tree in Figure 1, assume a query is issued at time $q = 6$. We follow the path $(v_6, v_4, v_8, \ldots)$. Along the path, $v_4$ is on the left, where $a, c \in D(v_4)$ are deleted after $q = 6$; $v_8$ is on the right, where $d \in D(v_8)$ begins at (or before) $q = 6$. Thus we report $D_6 = \{a, c, d\}$, which are exactly the elements present in the dataset at the query time.*

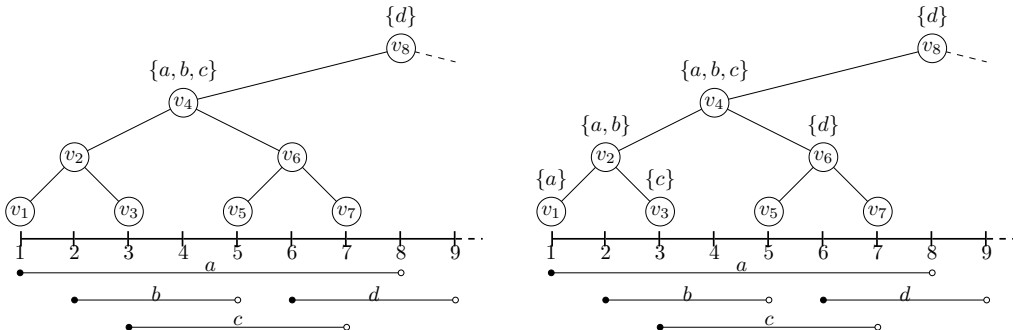

Figure 1: A standard interval tree.          Figure 2: An online interval tree.

In an online setting, however, we do not know which node is the highest to put an interval in, since we do not know the deletion time when an item is inserted. Consider timestamp 3 in Figure 1, where item $c$ is inserted into $D_2 = \{a, b\}$. If $c$ were to be deleted at timestamp 4, then $v_3$ becomes the highest node that stabs it, which by definition should store $c$. However, if $c$ is deleted at some timestamp among $\{5, 6, 7\}$, then $v_4$ is the highest node. Other possible candidates are $v_8, v_{16}, \ldots$. One idea is to put a copy of the interval into every node where the interval *might* be placed into. But there are infinitely many such nodes, therefore we do so lazily. More formally, we design a novel *online interval tree* that is capable of handling an infinite stream.

**Definition 5** (Online Interval Tree). *In an online interval tree $\mathcal{T}$, a tree node $v_t$ stores an interval $[i, j)$ if and only if both A) $i \leq t < j$; and B) $v_i$ is in the subtree rooted at $v_t$.*

Compared with the standard offline interval tree, an online interval tree may store an interval $[i, j)$ at multiple nodes. Nevertheless, condition B) implies that all these nodes lie on the root-to-node path to $v_i$, so at each level, there is at most one node that stores $[i, j)$, which is the key property we will need. We use Figure 2 to illustrate. Interval $a$ is stored at $v_1, v_2, v_4$, which are ancestors of $v_1$. It is not stored in $v_8$ since condition A) is violated: intuitively, by timestamp 8, $a$ is already deleted, so there is no need to store $a$ at $v_8$. On the other hand, $a$ needs to be stored in all $v_1, v_2, v_4$ (in the standard interval tree, it is only stored at $v_4$), because by timestamp 1 or 2, we still do not know the deletion time of $a$. Similarly, interval $d$ is stored in both $v_6$ and $v_8$.

### 3.1.1   Building the Online Interval Tree

This online interval tree can be incrementally constructed easily. After observing the update at timestamp $t$, we first compute the dataset $D_t$ at time $t$. These are exactly the elements that satisfy condition A). Then we construct the dataset $D(v_t)$ for the node $v_t$ out of elements in $D_t$, keeping only those that also satisfy condition B). These will be the intervals whose insertion-time node $v_i = v_t$ or lies in the left-subtree of $v_t$. For any node on the left-most path of $\mathcal{T}$ (where $t$ is a power of 2), $D(v_t) = D_t$ simply contains all elements in the current dataset. Otherwise, $D(v_t) \subseteq D_t$ will only contain items inserted after its closest left-ancestor.

**Example 2** (Online Interval Tree Construction). *In Figure 2, we construct $D(v_4) = D_4 = \{a, b, c\}$ at $t = 4$. To construct $D(v_6)$, we first compute $D_6 = \{a, c, d\}$. But we only consider items whose insertion-time is in the left-subtree (namely $\{v_5, v_6\}$), therefore $D(v_6) = \{d\}$ will only include $d$ from $D_6$. The intuition is that $a$ and $c$ have already been covered by $v_4$.*

Note that when $D(v_t)$ is first constructed, we do not have the deletion times of the items in $D(v_t)$, which will be added when these items are actually deleted later. For example, in Figure 2, $D(v_2) = \{a, b\}$ is constructed at timestamp 2 but neither item is associated with a deletion time. After timestamp 5, we add the deletion time of $b$, augmenting $D(v_2)$ to $\{a, (b, 5)\}$; after timestamp 8, it becomes $\{(a, 8), (b, 5)\}$. Note that there is no need to associate the left endpoints (i.e., insertion times) to the items as in the standard interval tree, and we will see why below.

### 3.1.2   Querying the Online Interval Tree

Now we show how to answer a stabbing query using the online interval tree. Since the online interval tree includes multiple copies of an item, the standard interval tree query algorithm will not work, as it may report duplicates. For the stabbing problem itself, duplicates are not an issue as they can be easily removed if they have been reported already. However, for answering linear queries, we actually need to cover all stabbed intervals by a disjoint union of subsets. To achieve it, we modify the stabbing query process as follows. Given a query at time $q$. We follow the root-to-node path to $v_q$ in $\mathcal{T}$, and only consider *left* nodes $v_l$ on the path where $l \le q$. For each $v_l$, we report all intervals $[i, j)$ in $D(v_l)$ where $j > q$.

**Example 3** (Online Interval Tree Query). *Again consider a query at $q = 6$ in Figure 2, on the root-to-node path, nodes $v_4$ and $v_6$ satisfy $l \le q$. We visit them and report $\{a, c\}$ and $\{d\}$ respectively, which jointly form the dataset $D_6$.*

Unlike in the standard interval tree, we do not query those *right* ancestors (e.g. $v_8$). It turns out that the items stored in the right ancestors are exactly compensated by the extra copies of items stored in the left ancestors of $v_q$. The following lemma formalizes this guarantee.

**Lemma 6.** *The query procedure described above reports each stabbed interval exactly once.*

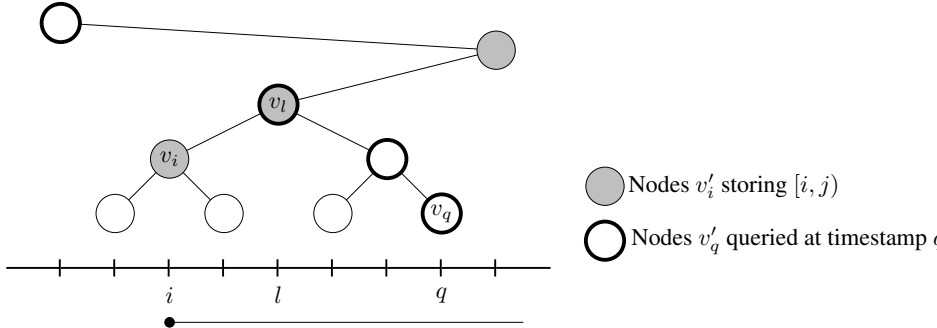

Figure 3: Querying a stabbed interval.

*Proof.* Given a query at time $q$, consider any item inserted at time $i$ and deleted at time $j$, represented by interval $[i, j)$. We first prove that this item will not be reported if it is not in the dataset $D_q$, i.e. interval $[i, j)$ is not stabbed by $q$. This happens when: (1) the item has been deleted at query time ($j \le q$). As we only report an item whose deletion time $j > q$, the interval is filtered out; (2) the item

has not arrived by query time ($i > q$). As we only visit *left* nodes $v_l$ where $l \leq q$, it follows that $l \leq q < i$. By definition, $v_l$ can store an item only if $l \geq i$, so this item is not stored by $v_l$.

The final case is when $i \leq q < j$, and the interval is supposed to be reported by exactly one node. This is shown in Figure 3. For the trivial case that $q = i$, the newly constructed node $v_i$ is the only node reporting this interval. Otherwise, consider the minimum subtree containing both $v_i$ and $v_q$, and assume it is rooted at $v_l$. We must have $v_i$ in its left subtree and $v_q$ in its right subtree by the minimum property, with the only exception that one of them can be $v_l$ itself, i.e., $i \leq l \leq q$. We can argue that $v_l$ is the only node that reports the interval: any node $v_i' \neq v_l$ that stores $[i, j)$ is either in the left subtree of $v_l$, or an ancestor of $v_l$ that is on the right side of $v_l$; any node $v_q' \neq v_l$ queried at $q$ is either in the right subtree of $v_l$, or an ancestor of $v_l$ that is on the left side of $v_l$. Thus the only node that can possibly report this interval is $v_l$. Since $v_l$ is queried and $j > q$, this stabbing interval is reported exactly once by $v_l$. □

## 3.2 Deletion-only Mechanism at Each Node

We have shown that the online interval tree can be incrementally constructed, such that at any time $t$, we can obtain the current dataset $D_t$ by a disjoint union of $O(\log t)$ subsets, each from $v_t$ or a left ancestor of $v_t$ in the interval tree. Consider each queried node $v_i$ ($i \leq t$) where $D_t(v_i)$ denotes the set of items that node $v_i$ stores at time $t$. This implies that a linear query $f(D_t)$ can be answered by computing the sum $\sum_i f(D_t(v_i))$ over queried nodes $v_i$, as specified in Section 3.1.2. Answering queries $\mathcal{F}$ on items stored by $v_i$ at time $t$ is a *deletion-only* problem: when $D(v_i)$ is first constructed at timestamp $i$, it has size $n(v_i) \leq n_i$ and no element is associated with deletion time. Then, items in $D(v_i)$ get deleted as time goes by.

Now we focus on the deletion-only problem at node $v_i$. To distinguish, we use $D(v_i)$ to denote the *initial state* of node $v_i$ when it first gets constructed at timestamp $i$, and use $D_t(v_i)$ to denote the *remaining items* in node $v_i$ at time $t$. Conceptually, we also consider a dynamic dataset $D_t^-(v_i)$, which consists of all the *deleted elements* from node $v_i$. Their sizes will be denoted as $n(v_i), n_t(v_i)$ and $n_t^-(v_i)$ respectively. For example, $D(v_2) = \{a, b\}$ when constructed at time 2, whereas $D_5(v_i) = \{a\}$ and $D_5^-(v_2) = \{b\}$. Clearly at any time $t$, $D_t(v_i) \uplus D_t^-(v_i) = D(v_i)$ and therefore $n_t(v_i) + n_t^-(v_i) = n(v_i)$. We then have $n_t(v_i) \leq n_t$ and $n_t(v_i) \leq n(v_i) \leq n_i$, but there is no relationship between $n_i$ and $n_t$.

A simple solution for this deletion-only problem is to first release a privatized $f(D(v_i))$ when initialized, and then run an insertion-only mechanism over the conceptual dataset $D_t^-(v_i)$ so that $f(D_t^-(v_i))$ can be obtained at any time. Using their difference, $f(D_t(v_i))$ can be answered. The problem here is that given a static mechanism $\mathcal{M}$ with error function $\alpha(|D|, \cdot)$, the initial $f(D(v_i))$ is answered with error $\alpha(n(v_i), \cdot)$. Although $n(v_i)$ is bounded by $n_i$, it has no relationship with the current data size $n_t$ at query time. In particular, the initial data size $n(v_i)$ can be arbitrarily larger than its current size $n_t(v_i)$, which fails to achieve our target error bound $\alpha(n_t, \cdot)$.

To fix the problem, we ensure that no more than $n(v_i)/2$ items should be deleted so that at any time we can guarantee $n(v_i) = O(n_t)$. When half the items have been deleted from $D(v_i)$, we restart the process on a new $D(v_i)$ that consists of the remaining items with fresh privacy budgets.

There are still a few privacy-related issues with the above idea. First, we cannot restart when exactly half the items have been deleted, which would violate DP. Instead, we run a basic counting mechanism $\mathcal{M}_{cnt,ins}$ over the conceptual dataset for deletions to approximately keep track of the number of deletions; we show that such an approximation will only contribute an additive polylogarithmic error. Second, since we restart the process above multiple times, we need to allocate the privacy budget carefully using sequential composition. But the privacy degradation is only polylogarithmic since we have only $O(\log n(v_i)) = O(\log t)$ restarts with high probability. Finally, as the decision of restarting the process depends on random noises, the total number of restarts can not be fixed in advance. Since we must guarantee differential privacy for any possible instantiation, we need to allocate the privacy budget through a convergent sequence. Similar to [4], we allocate the privacy budgets proportional to $r^{-(1+\eta)}$ in round $r$ for a small constant $\eta > 0$. The total privacy is then bounded regardless of the number of actual rounds, as $\sum_{r=1}^{\infty} r^{-(1+\eta)} < \frac{1}{\eta} + 1$. This incurs another logarithmic-factor degradation. Algorithm 1 details the steps we run at each node $v$ in the online

interval tree. We present in Lemma 7 its accuracy guarantee, assuming each node is allocated with $(\varepsilon, \delta)$-DP.

---

**Algorithm 1:** $(\varepsilon, \delta)$-DP Algorithm at node $v = v_i$

---

**Input:** Fully-dynamic stream $(\ldots, (s_t, x_t), \ldots)$, constant $\eta$, privacy budget $(\varepsilon, \delta)$, probability $\beta$, static ($\mathcal{M}_{\mathcal{F}}$) and insertion-only ($\mathcal{M}_{\mathcal{F},ins}$) mechanisms for queries $\mathcal{F}$, continual counting mechanism $\mathcal{M}_{cnt,ins}$

**Output:** $\mathcal{F}(D_t(v_i))$ at any time $t \geq i$

1   $\theta \leftarrow \frac{\eta}{1+\eta}, r \leftarrow 1, (\varepsilon_1, \delta_1) \leftarrow \left(\frac{\theta}{4}\varepsilon, \frac{\theta}{2}\delta\right), \beta_1 \leftarrow \frac{\theta}{6}\beta;$            // Initialize
2   $D(v_i) \leftarrow$ All items in $D_i$ inserted after the closest left-ancestor of $v_i$ in online interval tree;
3   $\boldsymbol{D} \leftarrow D(v_i);$
4   $\hat{\boldsymbol{n}} \leftarrow |\boldsymbol{D}| + \mathrm{Lap}(1/\varepsilon_1);$
5   **Release** $\mathcal{M}_{\mathcal{F}}(\boldsymbol{D})$ under $(\varepsilon_1, \delta_1)$-DP to answer $\mathcal{F}(D_i(v_i));$
6   Initiate $\mathcal{M}_{\mathcal{F},ins}$ under $(\varepsilon_1, \delta_1)$-DP and $\mathcal{M}_{cnt,ins}$ under $\varepsilon_1$-DP;

7   **foreach** $t \leftarrow i+1, i+2, \ldots$ **do**

8      **if** *update* $(s_t, x_t) = (-, x)$ *where* $x \in D(v_i)$ **then**
9         Augment the deletion time of $x$ to $(x, t);$
10        Feed an update $(+, x)$ to $\mathcal{M}_{\mathcal{F},ins}$ and $\mathcal{M}_{cnt,ins};$
11      **else**
12        Feed an update $(\perp, x)$ to $\mathcal{M}_{\mathcal{F},ins}$ and $\mathcal{M}_{cnt,ins};$

13      $\hat{\boldsymbol{n}}^- \leftarrow$ Approximate number of deletions from $\mathcal{M}_{cnt,ins};$
14      $\gamma \leftarrow$ Error bound $\gamma_{cnt}(t; \varepsilon_r, \beta_r)$ in Corollary 1;

15      **if** $\hat{\boldsymbol{n}}^- > \hat{\boldsymbol{n}}/2 + 2\gamma$ **then**            // Restart
16        $r \leftarrow r+1, (\varepsilon_r, \delta_r) \leftarrow \left(\frac{\theta}{4r^{1+\eta}}\varepsilon, \frac{\theta}{2r^{1+\eta}}\delta\right), \beta_r \leftarrow \frac{\theta}{6r^{1+\eta}}\beta;$
17        $\boldsymbol{D} \leftarrow D_t(v_i);$
18        $\hat{\boldsymbol{n}} \leftarrow |\boldsymbol{D}| + \mathrm{Lap}(1/\varepsilon_r);$

19        **if** $\hat{\boldsymbol{n}} < 2\gamma$ **then**            // Terminate
20           Halt by answering 0 to all future $\mathcal{F}(D_t(v_i));$

21        **Release** a new $\mathcal{M}_{\mathcal{F}}(\boldsymbol{D})$ under $(\varepsilon_r, \delta_r)$-DP to answer $\mathcal{F}(D_t(v_i));$
22        Restart $\mathcal{M}_{\mathcal{F},ins}$ under $(\varepsilon_r, \delta_r)$-DP and $\mathcal{M}_{cnt,ins}$ under $\varepsilon_r$-DP;
23      **else**
24        $\mathcal{F}(\boldsymbol{D}) \leftarrow$ Query the current $\mathcal{M}_{\mathcal{F}}(\boldsymbol{D});$
25        $\mathcal{F}(\boldsymbol{D}^-) \leftarrow$ Query the current $\mathcal{M}_{\mathcal{F},ins};$
26        Answer $\mathcal{F}(D_t(v_i)) \leftarrow \mathcal{F}(\boldsymbol{D}) - \mathcal{F}(\boldsymbol{D}_t^-);$

---

**Lemma 7.** *For each node $v_i$, Algorithm 1 is $(\varepsilon, \delta)$-DP. Suppose there is a static mechanism $\mathcal{M}_{\mathcal{F}}$ for answering query class $\mathcal{F}$ is equipped with error function $\alpha^{(k)}(\varepsilon, \delta, \beta, |D|, |\mathcal{F}|, |\mathcal{X}|)$. Then at any time $t$, Algorithm 1 answers $\mathcal{F}(D_t(v_i))$ with error $\alpha^{(\log t)}\left(\frac{\varepsilon}{\tilde{O}(1)}, \frac{\delta}{\tilde{O}(1)}, \beta, n_t + \tilde{O}(1), |\mathcal{F}|, |\mathcal{X}|\right).$*

*Proof.* We use some extra notation in the proof for simplicity. As we focus on the deletion-only problem at node $v_i$, we denote $\boldsymbol{D} = D_{t_0}(v_i)$, $\boldsymbol{D}^- = \boldsymbol{D} - D_t(v_i)$, $\boldsymbol{n} = n_{t_0}(v_i)$ and $\boldsymbol{n}^- = \boldsymbol{n} - n_t(v_i)$, where $t$ is the query time and $t_0$ is the beginning time of the current round. $\gamma$ is the public error bound of basic counting on infinite streams given in Corollary 1, which only depends on parameters in the round $r$ and the time $t$.

**Privacy.** Algorithm 1 uses four black-box mechanisms. In each round, $\hat{\boldsymbol{n}}$ is released using the Laplace mechanism and $\mathcal{F}(\boldsymbol{D})$ is released using static mechanism $\mathcal{M}_{\mathcal{F}}$. In addition, two insertion-only mechanisms $\mathcal{M}_{\mathcal{F},ins}$ and $\mathcal{M}_{cnt,ins}$ are used to track $\mathcal{F}$ and the basic counting query over the deletions $\boldsymbol{D}^-$. In any round $r$, the composition of these four mechanisms is $(4\varepsilon_r, 2\delta_r) = \left(\frac{\theta}{r^{1+\eta}}\varepsilon, \frac{\theta}{r^{1+\eta}}\delta\right)$-DP. As we restart these mechanisms, they are sequentially composed, which guarantees the whole mechanism at node $v_i$ is $(\varepsilon, \delta)$-DP, since $\sum_{r=1}^{\infty} \frac{\theta}{r^{1+\eta}} < \frac{\eta}{1+\eta}(\frac{1}{\eta} + 1) = 1$. Note that the privacy guarantee holds regardless of the number of restarts.

**Accuracy.** Let random variable $R$ denote the number of restarts before the mechanism terminates, we first bound $R$ as follows. When a restart happens, we have $\hat{\boldsymbol{n}}^- > \hat{\boldsymbol{n}}/2 + 2\gamma$, where $\gamma = \tilde{O}(1)$. With probability $1 - 2\beta_r$, both $\hat{\boldsymbol{n}}^-$ and $\hat{\boldsymbol{n}}/2$ have error at most $\gamma$. Conditioned on this happening, we get $\boldsymbol{n}^- > \boldsymbol{n}/2$: at least half of the remaining items have been deleted from $\boldsymbol{D}$. As $v_i$ was initialized with $n(v_i) \leq n_i \leq N_i \leq N_t$ items, with high probability this can happen at most $O(\log N_t)$ times before there are only $n_t(v_i) \leq \gamma$ items left. Still conditioned on the noise being small, line 19 becomes true and the algorithm halts by answering 0, which has error at most $O(\gamma)$ for any linear query. To conclude, conditioned on the events that in each round, the noises in counting are smaller than the bound $\gamma$, which happens with probability $1 - \sum_{r=1}^{\infty} 2\beta_r > 1 - \beta/3$, there can only be $O(\log N_t)$ rounds at time $t$, where $N_t$ is the number of updates up till time $t$.

We next bound the error of $\mathcal{F}(D_t(v_i))$ in any round $r \leq R$, as a function of $r$ and with probability $1 - 2\beta/3$. If the algorithm decides to restart at time $t$, an up-to-date dataset $D_t(v_i)$ is computed and a fresh $\mathcal{F}(D_t(v_i))$ is obtained from the static mechanism $\mathcal{M}_{\mathcal{F}}$ with privacy budget $(\varepsilon_r, \delta_r)$, whose error is $\alpha\left(\varepsilon_r, \delta_r, \frac{2\beta}{3}, n_t(v_i), \cdot\right) = \alpha(\varepsilon_r, \delta_r, \beta, n_t, \cdot)$ with probability $1 - 2\beta/3$. Otherwise (line 23), with probability $1 - 2\beta_r$ we have the actual number of deletions $\boldsymbol{n}^- \leq \boldsymbol{n}/2 + 4\gamma$. Namely the current data size is at least $n_t(v_i) = \boldsymbol{n} - \boldsymbol{n}^- \geq \boldsymbol{n}/2 - 4\gamma$. Conditioned on this, we calculate $\mathcal{F}(D_t(v_i))$ from the difference of $\mathcal{F}(\boldsymbol{D})$ and $\mathcal{F}(\boldsymbol{D}^-)$. Note that with probability $1 - \beta/6$, the error for $\mathcal{F}(\boldsymbol{D})$ from $\mathcal{M}_{\mathcal{F}}$ is $\alpha\left(\varepsilon_r, \delta_r, \frac{\beta}{6}, \boldsymbol{n}, \cdot\right)$ by definition; also with probability $1 - \beta/6$, the error for $\mathcal{F}(\boldsymbol{D}^-)$ from $\mathcal{M}_{\mathcal{F}, ins}$ is $\alpha^{(\log t)}\left(\frac{\varepsilon_r}{\log t}, \frac{\delta_r}{\log t}, \frac{\beta}{6}, \boldsymbol{n}^-, \cdot\right)$ by Lemma 4. But now both terms can be bounded by $\alpha^{(\log t)}\left(\frac{\varepsilon_r}{\log t}, \frac{\delta_r}{\log t}, \beta, n_t + \gamma, \cdot\right)$ with high probability, as $\boldsymbol{n}^- \leq \boldsymbol{n} \leq 2n_t(v_i) + 8\gamma = O(n_t + \gamma)$. This means their difference, $\mathcal{F}(D_t(v_i))$ has error $\alpha^{(\log t)}\left(\frac{\varepsilon_r}{\log t}, \frac{\delta_r}{\log t}, \beta, n_t + \gamma, \cdot\right)$ with probability at least $1 - 2\beta_r - 2\beta/6 \geq 1 - 2\beta/3$.

Finally we condition on the event that there are only $r = O(\log N_t)$ rounds to get rid of the dependency on $r$, which happens with all but $\beta/3$ probability. By Corollary 1, the counting error in round $r$ at time $t$ is $\gamma(t; \varepsilon_r, \beta_r) = O\left(\frac{\log^{1.5} t}{\varepsilon_r} \sqrt{\log \frac{1}{\beta_r}} + \frac{\log t}{\varepsilon_r} \cdot \log \frac{1}{\beta_r}\right) = O\left(\frac{\log^{1.5} t}{\varepsilon_r} \cdot \log \frac{1}{\beta_r}\right)$. Therefore the error for $\mathcal{F}(D_t(v_i))$ is (with probability $1 - \beta$)

$$\alpha^{(\log t)}\left(\frac{\varepsilon}{(\log^{1+\eta} N_t) \log t}, \frac{\delta}{(\log^{1+\eta} N_t) \log t}, \beta, n_t + \frac{(\log^{1+\eta} N_t) \log^{1.5} t}{\varepsilon} \log\left(\frac{\log N_t}{\beta}\right), |\mathcal{F}|, |\mathcal{X}|\right).$$

$\square$

### 3.3 Full Algorithm

Lemma 7 assumes each node is under $(\varepsilon, \delta)$-DP, which we cannot afford since we have an online interval tree of depth $\log t$. Since the tree size grows with $t$ and can be infinite, we allocate $(\varepsilon(v), \delta(v))$ proportional to $1/\ell^{1+\eta}$ to a node $v$ at level $\ell$ in the online interval tree, so that the composition of mechanisms at all the nodes still satisfies $(\varepsilon, \delta)$-DP at any time $t$. Since each item will only affect one single node in each level, nodes at the same level enjoy parallel composition. The error of the final sum is then decomposed into $\log t$ online interval tree nodes, where the error of each node is given by Lemma 7 but under $(\varepsilon(v), \delta(v))$-DP.

To provide an error guarantee to the final results, note that it is the disjoint union of $(\log^2 t)$ static mechanisms. This is because each of the $(\log t)$ queried online interval tree nodes runs an insertion-only mechanism to support querying the deleted elements, the error of each one is from the error of $(\log t)$ static mechanisms. We are left with analyzing the error of each building block: the static mechanisms. Their error depends on the privacy budgets $(\varepsilon', \delta')$ allocated to it. In our construction, it has a polylogarithmic degradation compared to the $(\varepsilon, \delta)$ for the whole mechanism. In particular, there are 3 factors that accounts for the allocation of privacy budgets.

1. We have an online interval tree of height $(\log t)$. While nodes on the same level enjoy parallel composition, it is possible that the change of one timestamp affects multiple tree nodes on the root-to-node path corresponding to this timestamp. Therefore, we must allocate privacy budget with sequential composition. Moreover, the tree can grow infinitely. So

instead of an even allocation, we apply a convergent sequence to divide the budgets, which causes a $\log^{1+\eta} t$ overhead to the worst building block at time $t$.

2. At each online interval tree node, we restart the 4 mechanisms several times. Since this may repeatedly reveal information of the same entry, we also need to divide the privacy budget accordingly. Again, we allocate the budgets in each round using a convergent sequence again to make the DP guarantee independent of the number of restarts. With high probability, no queried node will restart more than $\log N_t$ times, so the minimum privacy budget in any round is a $1/(\log^{1+\eta} N_t)$ fraction of the budget allocated to the node.

3. Finally, in each round of each node, the insertion-only mechanisms $\mathcal{M}_{cnt,ins}$ and $\mathcal{M}_{\mathcal{F},ins}$ are built from static mechanisms, each receiving a $1/(\log t)$ fraction of the budget.

Putting things together, we arrive at the main result of this paper. For the running time, observe that the mechanism run at each node of $\mathcal{T}$ is dominated by the insertion-only mechanism from Lemma 4 that tracks the deletions. Since there can be $O(\log t)$ restarts, and the total running time over all nodes in the same level is $O(\log t) \cdot \text{time}(N_t)$. Summing over all $O(\log t)$ levels, we obtain the running time stated.

**Theorem 8.** *If there is a static DP mechanism with error function $\alpha^{(k)}(\varepsilon, \delta, \beta, |D|, |\mathcal{F}|, |\mathcal{X}|)$ for a set of union-preserving queries $\mathcal{F}$, then there is an $(\varepsilon, \delta)$-DP mechanism $\mathcal{M}_{dyn}$ for fully-dynamic streams, so that at any time $t$, it answers queries $\mathcal{F}$ on $D_t$ with error*

$$\zeta(t; \varepsilon, \delta, \beta, n_t, |\mathcal{F}|, |\mathcal{X}|) = \alpha^{(\log^2 t)}\left(\frac{\varepsilon}{\log^{3+2\eta} t}, \frac{\delta}{\log^{3+2\eta} t}, \beta, n_t + \frac{\log^{3.5+2\eta} t}{\varepsilon}\log\left(\frac{\log t}{\beta}\right), |\mathcal{F}|, |\mathcal{X}|\right),$$

*for any constant $\eta > 0$. The running time of $\mathcal{M}_{dyn}$ up to time $t$ is $O(\log^2 t) \cdot \text{time}(N_t)$.*

Plugging in PMW as the static DP mechanism, we obtain a fully dynamic algorithm for answering a set of arbitrary linear queries with the following error bounds:

**Corollary 2.** *Given a set of linear queries $\mathcal{F}$, there is an $(\varepsilon, \delta)$-DP mechanism $\mathcal{M}_{\mathcal{F},dyn}$ that at any time $t$, answers any query $f \in \mathcal{F}$ on a fully-dynamic stream with error*

$$\zeta_{\mathcal{F}}(t; \varepsilon, \delta, \beta, n_t, |\mathcal{F}|, |\mathcal{X}|) = \alpha_{\text{PMW}}^{(\tilde{O}(1))}\left(\frac{\varepsilon}{\tilde{O}(1)}, \frac{\delta}{\tilde{O}(1)}, \beta, n_t + \tilde{O}(1), |\mathcal{F}|, |\mathcal{X}|\right) = \begin{cases} \tilde{O}(\sqrt{n_t}), & \delta > 0, \\ \tilde{O}(n_t^{2/3}), & \delta = 0. \end{cases}$$

Our mechanism only has a polynomial dependency on the data size $n_t$ at time $t$, matching results in the static setting, whereas a straightforward extension of insertion-only mechanisms [9, 4] will have a polynomial dependency on the number of updates $N_t \gg n_t$.

## 4    Limitations

While polylogarithmic factors are often neglected in theoretical studies, they still present significant overhead in practice, which limits the practicality of the algorithms proposed in this paper. How to reduce this overhead remains an interesting problem for further investigation. A possible future direction is to consider white-box mechanisms that improve the accuracy (though they can only reduce the polylogarithmic factors) while being practical. It is also interesting to study how to answer non-union-preserving queries (e.g. distinct count) accurately under the streaming DP setting.

### Acknowledgments and Disclosure of Funding

This work is supported by HKRGC under grants 16205422, 16204223, and 16203924; and by the Start-up Research Fund of Southeast University under grant RF1028625150. We would also like to thank the anonymous reviewers who have made valuable suggestions on improving the presentation of the paper.

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

# A DP Mechanisms for Linear Queries: A Review

In this section, we present some important DP mechanisms for linear queries and their error bounds. The analysis is similar to that in in [30], but we clarify the dependency on $\beta$, which will be needed for analyzing the simultaneous error in the streaming setting. Note that a linear query has sensitivity 1 by definition.

**Laplace Mechanism.** When $\mathcal{F} = \{f\}$ is a single query, the Laplace mechanism that outputs $\mathcal{M}_{\mathrm{Lap}}(D) = f(D) + \mathrm{Lap}(\frac{1}{\varepsilon})$ has error $\alpha_{\mathrm{Lap}}(\varepsilon, \beta) = \frac{1}{\varepsilon} \ln \frac{1}{\beta}$. When $\mathcal{F}$ contains multiple queries, we may add $\mathrm{Lap}(\frac{|\mathcal{F}|}{\varepsilon})$ noise to each query result and apply basic composition to guarantee $\varepsilon$-DP of the whole mechanism. To translate it into an error bound, we bound the failure probability of each noise by $\frac{\beta}{|\mathcal{F}|}$, so that a union bound will bring the total failure probability to $\beta$. A similar argument can be made using advanced composition. To conclude, answering a set of queries $\mathcal{F}$ using Laplace mechanism achieves error (for $\delta \geq 0$)

$$
\alpha_{\mathrm{Lap}}(\varepsilon, \delta, \beta, |\mathcal{F}|) = \begin{cases} O\left(\dfrac{|\mathcal{F}|}{\varepsilon} \log \dfrac{|\mathcal{F}|}{\beta}\right) & , \delta \leq e^{-\Omega(|\mathcal{F}|)}\,; \\[3mm] O\left(\dfrac{\sqrt{|\mathcal{F}| \log \frac{1}{\delta}}}{\varepsilon} \log \dfrac{|\mathcal{F}|}{\beta}\right) & , \delta \geq e^{-O(|\mathcal{F}|)}\,. \end{cases}
$$

**Gaussian Mechanism.** Similar to the Laplace mechanism, the Gaussian mechanism protects $(\varepsilon, \delta)$-DP of query $f$ by outputting $\mathcal{M}_{\mathrm{Gauss}}(D) = f(D) + \mathcal{N}\left(0, \frac{2}{\varepsilon^2} \ln \frac{1.25}{\delta}\right)$. Its error is $\alpha_{\mathrm{Gauss}}(\varepsilon, \delta, \beta) = \frac{2}{\varepsilon} \sqrt{\ln \frac{1.25}{\delta} \ln \frac{2}{\beta}}$. When composing multiple Gaussian mechanisms that each answers a query from $\mathcal{F}$, zCDP composition [2] can be applied, which shows adding $\mathcal{N}\left(0, O\left(\frac{|\mathcal{F}|}{\varepsilon^2} \log \frac{1}{\delta}\right)\right)$ noise to each query suffices to protect $(\varepsilon, \delta)$-DP of the whole mechanism. Therefore answering a set of queries $\mathcal{F}$ using Gaussian mechanism achieves error (for $\delta > 0$)

$$
\alpha_{\mathrm{Gauss}}(\varepsilon, \delta, \beta, |\mathcal{F}|) = O\left(\frac{\sqrt{|\mathcal{F}| \log \frac{1}{\delta}}}{\varepsilon} \sqrt{\log \frac{|\mathcal{F}|}{\beta}}\right)\,.
$$

**Private Multiplicative Weights.** When there are many queries $|\mathcal{F}| = \Omega(|D|)$, composing individual mechanisms has error polynomial in $|\mathcal{F}|$, thus also in $|D|$. The Private Multiplicative Weights mechanism [15, 17] performs better in this case. The following error bound is presented in [17, 8].

$$
\alpha_{\mathrm{PMW}} = \begin{cases} O\left(\left(\dfrac{|D|^2 \log |\mathcal{X}| \log(|\mathcal{F}|/\beta)}{\varepsilon}\right)^{\frac{1}{3}}\right) & , \delta = 0\,; \\[4mm] O\left(\left(\dfrac{|D| \sqrt{\log |\mathcal{X}| \log(1/\delta)} \log(|\mathcal{F}|/\beta)}{\varepsilon}\right)^{\frac{1}{2}}\right) & , \delta > 0\,. \end{cases}
$$

Apart from mechanisms mentioned above, there are other private mechanisms for linear queries. For example, the optimal composition [21] can be used in place of basic or advanced composition to provide a better allocation of privacy budget, yet computing it is costly. The $\log |\mathcal{F}|$ factor is removable for Laplace mechanism [29] and almost removable for Gaussian mechanism [14]. Under pure-DP, SmallDB [1] has asymptotically the same error as PMW, but its running time is prohibitive. The Matrix mechanism [22, 23] exploits structural properties within the query set $\mathcal{F}$ and works well in practice. But it does not have a closed-form error bound for general queries, and finding an optimal querying strategy is time-consuming.

In general, the best mechanism is related to the hereditary discrepancy [16, 26] of the set of queries. For example, for $d$-dimensional halfspace counting queries, [26] has error $\tilde{O}(|D|^{\frac{1}{2} - \frac{1}{2d}})$. In this paper we use $\alpha$ as a function of $\varepsilon, \delta, \beta$, and possibly $|D|, |\mathcal{F}|, |\mathcal{X}|$ to denote the error of any mechanism answering linear queries on *static* datasets, without detailing the best mechanism under a specific setting and choice of the parameters. Since our paper takes a black-box approach, all these algorithms can be plugged into our framework so as to support dynamic data, while incurring a polylogarithmic-factor degradation.

# B  Error Bounds under Disjoint Union

Note that $\alpha^{(k)}$ denotes the error of the sum of $k$ mechanisms, each of which having error $\alpha$ under the same parameter settings. It always holds that $\alpha^{(k)}(\cdot, \beta) \leq k \cdot \alpha(\cdot, \frac{\beta}{k})$ by union bound. In this section, we show cases where $\alpha^{(k)}$ can be tightened for specific mechanisms. As our running example, consider $\alpha_{\text{Lap}}(\varepsilon, \beta) = \frac{1}{\varepsilon} \ln \frac{1}{\beta}$. The union bound reduction above gives

$$\alpha_{\text{Lap}}^{(k)}(\varepsilon, \beta) \leq \frac{k}{\varepsilon} \ln \frac{k}{\beta}$$

to bound the error of summing $k$ Laplace noises $\text{Lap}(\frac{1}{\varepsilon})$. Next we show how this can be tightened.

## B.1  Unbiasedness

If a mechanism $\mathcal{M}$ is *unbiased* with error $\alpha$, naturally the error only scales with $\sqrt{k}$. To see why this is true, we can argue that with all but $\frac{\beta}{2}$ probability, individual mechanisms have their errors bounded by $\alpha(\cdot, \frac{\beta}{2k})$ simultaneously. Conditioned on this happening, apply Hoeffding's inequality with the remaining $\frac{\beta}{2}$ probability, we get

$$\alpha_{\text{Unbiased}}^{(k)}(\cdot, \beta) \leq \sqrt{2k \ln \frac{4}{\beta}} \cdot \alpha\left(\cdot, \frac{\beta}{2k}\right) .$$

This can be applied to the Laplace mechanism to get

$$\alpha_{\text{Lap}}^{(k)}(\varepsilon, \beta) = O\left(\frac{\sqrt{k}}{\varepsilon} \sqrt{\log \frac{1}{\beta}} \log \frac{k}{\beta}\right) .$$

## B.2  Concentration Bounds

For specific distributions like Laplace (sub-exponential) and Gaussian (sub-gaussian), the concentration bounds are usually tighter than using unbiasedness only. It helps save the $\log k$ factor from union bound. Consider the Laplace mechanism, note that the $\text{Lap}(\frac{1}{\varepsilon})$ random variable is sub-exponential with norm $\|\text{Lap}(\frac{1}{\varepsilon})\|_{\Psi_1} = \frac{2}{\varepsilon}$. We can then apply Bernstein's inequality [31].

**Lemma 9** (Bernstein's inequality). *Let $X_1, \ldots, X_k$ be i.i.d. zero-mean sub-exponential random variables with norm $\Psi_1$. There is an absolute constant $c$ so that for any $t \geq 0$,*

$$\Pr\left[\left|\sum_{i=1}^{k} X_i\right| > t\right] \leq 2 \exp\left[-c \min\left\{\frac{t^2}{k\Psi_1^2}, \frac{t}{\Psi_1}\right\}\right] .$$

We therefore conclude the Laplace mechanism has error function

$$\alpha_{\text{Lap}}^{(k)}(\varepsilon, \beta) = O\left(\frac{1}{\varepsilon} \sqrt{\left(k + \log \frac{1}{\beta}\right) \cdot \log \frac{1}{\beta}}\right) . \tag{2}$$

As another example, for the Gaussian mechanism, the sum of $k$ Gaussian noises is still a Gaussian noise with its variance scaled up by $k$, thus the disjoint union of $k$ Gaussian mechanisms has error

$$\alpha_{\text{Gauss}}^{(k)}(\varepsilon, \delta, \beta) = O\left(\frac{1}{\varepsilon} \sqrt{k \log \frac{1}{\delta} \log \frac{1}{\beta}}\right) .$$

For the PMW mechanism, the union bound remains the best we know, that is

$$
\alpha_{\text{PMW}}^{(k)} =
\begin{cases}
O\left(k \cdot \left(\dfrac{|D|^2 \log |\mathcal{X}| \log(k|\mathcal{F}|/\beta)}{\varepsilon}\right)^{\frac{1}{3}}\right) & , \delta = 0 \, ; \\[2em]
O\left(k \cdot \left(\dfrac{|D|\sqrt{\log |\mathcal{X}| \log(1/\delta)} \log(k|\mathcal{F}|/\beta)}{\varepsilon}\right)^{\frac{1}{2}}\right) & , \delta > 0 \, .
\end{cases}
$$
$$
=
\begin{cases}
\tilde{O}\left(k \cdot |D|^{\frac{2}{3}}\right) & , \delta = 0 \, ; \\[1em]
\tilde{O}\left(k \cdot |D|^{\frac{1}{2}}\right) & , \delta > 0 \, .
\end{cases}
$$

## C   Related Work for DP under Continual Observation

In the Binary Tree mechanism [9], each timestamp is treated as a leaf node in the binary tree, and the mechanism privately releases the count of each tree node. If the stream has bounded length $T$, then each node receives a $\frac{1}{\log T}$ fraction of the privacy budget, and the final count consists of counts from $O(\log t)$ tree nodes. Using our notation, the mechanism has error $\alpha_{\text{Lap}}^{(\log t)}(\frac{\varepsilon}{\log T}, \beta)$ for basic counting. Chan et al. [4] first showed that allocating the privacy budget through a convergent sequence will allow the Binary Tree mechanism to also work for infinite streams, where the error is worse by a logarithmic factor as $\alpha_{\text{Lap}}^{(\log t)}(\frac{\varepsilon}{\log^{1+\eta} t}, \beta)$ for constant $\eta > 0$. They further proposed a hybrid mechanism to show that $\alpha_{\text{Lap}}^{(\log t)}(\frac{\varepsilon}{\log t}, \beta)$ can be achieved by separately releasing the counts for $[1, 2), [2, 4), [4, 8), \ldots$. Using the $\alpha_{\text{Lap}}^{(k)}$ function in Equation (2) above, we obtain the error

$$
\alpha_{\text{Lap}}^{(\log t)}\left(\frac{\varepsilon}{\log t}, \beta\right) = O\left(\frac{\log^{1.5} t}{\varepsilon} \sqrt{\log \frac{1}{\beta}} + \frac{\log t}{\varepsilon} \cdot \log \frac{1}{\beta}\right) .
$$

for continual counting, which was presented in Corollary 1 as our building block. Note that this is a *per-query* bound: with constant probability, a query at any time $t$ can be answered with error $O(\frac{1}{\varepsilon} \log^{1.5} t)$.

Alternatively, if we replace $\beta$ with $\beta/T$ where $T$ is an upper bound on $t$, we obtain a bound that holds *simultaneously* for all queries $1 \leq t \leq T$: with constant probability, any query can be answered with error $O(\frac{1}{\varepsilon} \log^2 T)$. The *simultaneous* bound given in [9] was $O(\frac{1}{\varepsilon} \log^{2.5} T)$ using unbiasedness but not concentration, which was corrected in [11]. In [4], Chan et al. presented the $\alpha^{(k)}$ function as $\alpha_{\text{Lap}}^{(k)}(\varepsilon, \beta) = O\left(\frac{\sqrt{k}}{\varepsilon} \log \frac{1}{\beta}\right)$, and thus

$$
\alpha_{\text{Lap}}^{(\log t)}\left(\frac{\varepsilon}{\log t}, \beta\right) = O\left(\frac{\log^{1.5} t}{\varepsilon} \log \frac{1}{\beta}\right) .
$$

While the per-query bound is the same, for the simultaneous bound, this only gives $O(\frac{1}{\varepsilon} \log^{2.5} T)$, whereas our tighter bound gives $O(\frac{1}{\varepsilon} \log^2 T)$. The same simultaneous bound was presented in [13, 18] with improved constants using the Matrix Mechanism [22, 23] as building block. For sparse finite streams, [11] achieves $O(\frac{\log^{1.5} N_t}{\varepsilon} \log \frac{1}{\beta} + \frac{1}{\varepsilon} \log \frac{T}{\beta})$ per-query error for basic counting, which has an asymptotic improvement when $N_t = t^{o(1)}$.

We briefly mention some other work under similar settings but are less related. [3] studies the DP histogram problem in the streaming setting, which is a special case of linear queries. Their main contribution is when the universe $\mathcal{X}$ or the sensitivity of the queries are unbounded. Otherwise, the algorithm is equivalent to Corollary 1. [19] studies the DP distinct counting problem under the turnstile (fully-dynamic) model. Distinct counting queries are known to be non-linear and not union-preserving, which is separate from the interest of this paper. There are also existing work that study graph queries under the continual observation model of differential privacy, e.g. [28, 12, 20, 27].

