# OpenReview forum: "Differential Privacy on Fully Dynamic Streams"
_NeurIPS.cc/2025/Conference — NeurIPS 2025 spotlight_

### Official Review · Reviewer_kHdG · 2025-06-20

**Clarity:** 4
**Significance:** 4
**Originality:** 3
**Rating:** 5
**Confidence:** 4

**Summary:**

This paper is about answering linear queries on a fully dynamic stream while preserving differential privacy. Previous work has only worked on insertion only streams where an element is never deleted. Moreover, the insertion only algorithms can be extended to fully dynamic but with a rather high error: their error is a function of $|D_t|$, the size of the dataset at time t. In a fully dynamic stream, if the number of elements ever inserted until time t is $N_t$, naively using insertion only algorithms gives an error that is a function of $N_t$ and not $|D_t|$, while $|D_t|$ can be much smaller than $N_t$.

This work shows how to use an insertion only algorithm in a blackbox way and get same errors in terms of the size of the dataset at time t. Their approach consists of many interesting components. First they introduce a dynamic interval tree that can cope with deletions and any query can be answered by only inspecting log t nodes. If $D^+_t$ is the set of nodes ever inserted up to time t and $D^-_t$ is the set of nodes deleted on or before t, they compute a linear query by computing it on $D^+_t$ minus its value on $D^-_t$. Their second component is "restarts", where they compute a query from scratch on $D_t$ using a static algorithm instead of computing it on  $D^+_t$ and  $D^-_t$. They do these restarts when almost half of the elements in a tree node are deleted, and this way they can control the error to be in a function of $|D_t|$. Their third component is a clever distribution of privacy budget which is not novel on its own but is nicely used.

**Questions:**

On line 341 you allocate privacy budget of $1/\ell^{1+\eta}$ to nodes at level $\ell$. However as time goes on the level of nodes can increase. Could you explain why this is fine?

**Ethical Concerns:**

["NO or VERY MINOR ethics concerns only"]

**Final Justification:**

This paper is strong and original enough for acceptance.

**Limitations:**

I can't think of any limitations other than the ones stated.

**Quality:**

4

**Strengths And Weaknesses:**

Strengths:
- The paper is very well written. They provide really good examples to explain their ideas, and as explained above I like the components of their algorithm. their contribution is strong.

Weaknesses:
- minor: the proof of Lemma 7 refers to Algorithm 1 which is stated after this lemma in the appendix. It would have be good to provide a summary of this algorithm or explain the parameters before referring to them.

---

> ### Author Rebuttal · Authors · 2025-07-26
>
> > W1. The proof of Lemma 7 refers to Algorithm 1 which is stated after this lemma in the appendix. It would have be good to provide a summary of this algorithm or explain the parameters before referring to them.
>
> We will add some description to Algorithm 1 in the main text.
>
> > Q1. On line 341 you allocate privacy budget of $1/\ell^{1+\eta}$ to nodes at level $\ell$. However as time goes on the level of nodes can increase. Could you explain why this is fine?
>
> The height of the tree is $\ell=O(\log t)$, so this only causes a polylogrithmic factor increase to the error, which is often neglected in theoretical studies.

---

### Official Review · Reviewer_yUVf · 2025-06-25

**Clarity:** 4
**Significance:** 3
**Originality:** 4
**Rating:** 5
**Confidence:** 3

**Summary:**

This paper studies privately answering linear queries that are union-preserving (i.e., $f(D_1\cup D_2)=f(D_1)+f(D_2)$) in the turnstile streaming model. Prior works mainly focus on insertion-only streams, as most of the DP tools can be naturally attuned in this model. Note that simply treats a turnstile stream as two insertion-only streams is not enough, as popular private mechanisms have square root dependencies on the size of the set, doing so would lead to undesirable bound on the error. Thus, to obtain reasonable error bound on turnstile streams, the authors use a novel online interval tree to keep track of the lifespan of elements. This is significantly different from classical interval trees, where all intervals are known in advance. Authors show that it is possible to maintain the intervals in an online fashion while not blowing up the complexity of the data structure significantly. Using this data structure, authors show that the problem is then reduced to a deletion-only problem, but running an insertion-only algorithm is not enough, so the proposed algorithm restarts the data structure when half of the elements are deleted. With some extra caveats being taken care of, the authors arrive at a private algorithm for linear queries over turnstile streams.

**Questions:**

1. What do you think are some other applications of the online interval tree when used for turnstile streams?

2. For non-union-preserving linear queries, is it possible to extend the developed techniques to handle them? If not, what are the major barriers?

**Ethical Concerns:**

["NO or VERY MINOR ethics concerns only"]

**Final Justification:**

The paper is very solid, I only have minor suggestions and authors have addressed them well.

**Limitations:**

Yes.

**Paper Formatting Concerns:**

N/A.

**Quality:**

4

**Strengths And Weaknesses:**

Strengths:

1. The result obtained in this paper is strong, it achieves nearly-optimal dependence (up to polylog factors) on the size of the stream, which is nontrivial. Although the core mechanism (private multiplicative weights and insertion-only algorithm) has been developed, designing the correct data structure to utilize these tools is an important contribution.

2. The online interval tree is a novel contribution that might find applications other than DP in the turnstile streaming setting. In fact, adapting the online interval tree for turnstile streams could be abstracted as a framework for tasks that have good insertion-only and static algorithms that depend on the size of the stream.

3. The paper overall is well-written, the algorithm is explained clearly and the online interval tree data structure, facilitated with several figures, is easy to understand.

Weaknesses:

1. The mechanism is developed for the general setting, it would be interesting to see more white-box application of the proposed algorithm for particular problems, such as half-space queries, with even more improved error bounds.

2. The paper is technically dense, thus the discussions on future directions are limited.

---

> ### Author Rebuttal · Authors · 2025-07-26
>
> > W1. The mechanism is developed for the general setting, it would be interesting to see more white-box application of the proposed algorithm for particular problems, such as half-space queries, with even more improved error bounds.
>
> > W2. The paper is technically dense, thus the discussions on future directions are limited.
>
> Our paper takes a black-box approach. We believe designing white-box mechanisms to improve the bounds is an interesting future direction.
>
> > Q1. What do you think are some other applications of the online interval tree when used for turnstile streams?
>
> We think the data structure will be useful for answering interval-stabbing queries and range queries in the online setting.
>
> > Q2. For non-union-preserving linear queries, is it possible to extend the developed techniques to handle them? If not, what are the major barriers?
>
> The problem for non-union-preserving queries is challenging since the tree-decomposition does not work. For example, if the dataset is divided into two subsets, each containing 40 and 50 distinct elements respectively, then the distinct count of the original dataset can be any value between 50 and 90.

---

> > ### Comment · Reviewer_yUVf · 2025-08-01
> >
> > I thank the authors for the response, though they seem to not address the weakness regarding discussions on future directions. I would suggest to include such discussions in the appendix.

---

> > > ### Author Response · Authors · 2025-08-02
> > >
> > > Thank you for the suggestion. Yes, we will include the discussions on future directions in the appendix, such as considering white-box reductions (though this can only improve the accuracy by polylog factors), making the algorithm more practical, and designing mechanisms for non-union-preserving queries, etc.

---

> > > > ### Comment · Reviewer_yUVf · 2025-08-02
> > > >
> > > > Thanks, I have no further questions, and will keep my score as is.

---

### Official Review · Reviewer_KNz3 · 2025-06-30

**Clarity:** 2
**Significance:** 4
**Originality:** 3
**Rating:** 5
**Confidence:** 3

**Summary:**

The paper tackles continual release of differentially-private answers for fully dynamic (turnstile) data streams, where elements may be both inserted and deleted. While previous work handled insertion-only streams or the static setting, the authors design a black-box transformation that turns any static DP mechanism for a union-preserving family of queries (e.g. linear queries) into a mechanism that operates on fully dynamic streams with only a polylogarithmic loss in accuracy and running-time.

The core technical contribution is an online interval tree that stores each item in at most one node per tree level, permitting a decomposition of the data set at time $t$ into $O(\log t)$ disjoint parts that can each be handled by an insertion-only sub-mechanism .

A careful choice of parameters bounds the cumulative privacy loss while keeping the error blow-up within polylogarithmic factors.
In particular, plugging private multiplicative weights as the static algorithm, they achieve $\tilde O(\sqrt{|D_t|})$ error for approximate DP and $\tilde O(|D_t|^{3/4})$ for pure-DP. Here $|D_t|$ is the size of the current dataset $D_t$. Notably, there is no dependence on the number of queries for linear queries, whereas straightforward extensions of prior insertion-only works would result in polynomial dependence on the number of queries.

**Questions:**

1. Are there any interesting classes of union-preserving query families beyond linear queries?

---

### Suggestions
1. I would suggest (informally) stating the main results within the introduction of the paper and quantifying the improvements from prior works.
2. There are some references for continual release which may be relevant.

---

- Song, Little, Mehta, Vinterbo, Chaudhuri. "Differentially private continual release of graph statistics." arXiv 2018.
- Fichtenberger, Henzinger, Ost. "Differentially Private Algorithms for Graphs Under Continual Observation." ESA 2021.
- Epasto, Liu, Mukherjee, Zhou. "Sublinear Space Graph Algorithms in the Continual Release Model." arXiv 2024.
- Jain, Smith, Wagaman. "Time-aware projections: Truly node-private graph statistics under continual observation." SP 2024.
- Raskhodnikova, Steiner. "Fully Dynamic Algorithms for Graph Databases with Edge Differential Privacy." PACMMoD 2025

**Ethical Concerns:**

["NO or VERY MINOR ethics concerns only"]

**Final Justification:**

The rebuttal clarified the scope of the contribution, specifically about union-preserving families. I think this will be a good contribution to NeurIPS.

**Limitations:**

yes

**Quality:**

3

**Strengths And Weaknesses:**

### Strengths
1. To the best of my knowledge, this is the first black-box method that matches static-query accuracy up to polylog factors for fully dynamic streams, closing the gap with insertion-only results.
1. The online interval tree data structure seems to be novel and may be of independent interest even beyond privacy.

---

### Weaknesses
1. The results only hold for union-preserving query families. While this includes the class of linear queries, it limits applicability.

---

> ### Author Rebuttal · Authors · 2025-07-26
>
> > W1. The results only hold for union-preserving query families. While this includes the class of linear queries, it limits applicability.
>
> The class of linear queries is extensively studied under differential privacy (e.g. references in line 97). This is a large family and serves as building block for many problems:
> - histograms, range counting, k-marginals and data cubes are well-known linear queries;
> - the sum/mean-estimation query $\sum_i x_i$ can be rewritten as $B\cdot \sum_i (x_i/B)$ where the weight for a value $v$ is $v/B$, assuming each item is bounded by $[0,B]$;
> - the sum-of-square query $\sum_i x_i^2$ for computing variance can be rewritten similarly;
> - the median query itself is non-linear (and not union-preserving), but it is known that any quantile query can be reduced to $(\log B)$ linear queries under DP.
>
> > Q1. Are there any interesting classes of union-preserving query families beyond linear queries?
>
> In fact, the definition for union-preserving queries (footnote 2 on page 2) can be generalized to any $+$ operator that is associative. For example, consider the truncated-linear query $g(D)=\min(f(D),\tau)$ where $f$ is a linear query and $\tau$ is a constant.  This non-linear query is also union-preserving by defining $a+b := \min(a+b,\tau)$.  Note that our error bound then also becomes the "sum" of the errors from a polylogarithmic number of the black-box static estimators, where the overloaded $+$ is used to compute the sum.
>
> > S1. I would suggest (informally) stating the main results within the introduction of the paper and quantifying the improvements from prior works.
>
> We have stated the main result near the end of introduction (line 57). We will emphasize it by adding a paragraph header in bold.
>
> > S2. There are some references for continual release which may be relevant.
>
> Thanks for the pointers.  We will include these references.

---

> > ### Comment · Reviewer_KNz3 · 2025-08-02
> >
> > Thanks for the clarification. I think this is a nice contribution if the authors can include the discussion in the rebuttal. I will maintain my score.

---

### Official Review · Reviewer_QjJv · 2025-06-30

**Clarity:** 3
**Significance:** 3
**Originality:** 3
**Rating:** 5
**Confidence:** 3

**Summary:**

While differential privacy (DP) oftentimes imposes a challenging trade-off between utility and privacy in the static setting, dynamic algorithms add another factor tearing this trade off apart, which is time (or number of updates). The authors of the submission present a construction for a dynamic, $(\epsilon, \delta)$-DP algorithm for any union-preserving query (e.g., linear queries), given a static, $(\epsilon', \delta')$-DP algorithm. A query $f$ is union preserving on multisets $D_1, D_2$ if $f(D_1 \cup D_2) = f(D_1) + f(D_2)$. For linear queries, using the DP multiplicative weights mechanism, their fully dynamic algorithm achieves an error of $\tilde{O}(n_t^{2/3})$ for pure DP ($\delta=0$), and $\tilde{O}(\sqrt{n_t})$ for approximate DP with a multiplicative $O(\log^2 t)$ blow-up in time, where $n_t$ is the size of the dynamic dataset at time $t$. This improves over existing constructions with similar error bounds but which are only partially dynamic (insertions or deletions), and fully dynamic constructions with error bounds that that are linear in $t$. For union-preserving queries, the bound is a bit more complex: Given a static DP mechanism with an of error $\alpha^{(k)}$ when transformed into an insert-only mechanism via appropriate standard techniques, roughly speaking, the resulting dynamic error bound is $\alpha^{(\log^2 t)}$, and dependencies on $\epsilon, \delta$, size of the dataset are substituted by themselves multiplied by a $O(\log^3 t)$ or $O(1/\log^3 t)$ factor. The running time is claimed as the running time of the static black-box algorithm on $n_t$ times $O(\log^2 t)$.

**Questions:**

The running time as stated is a bit confusing to me. It seems to me that it should be the time to run the static mechanism for every time step, plus the time for running the incremental mechanisms for the respective restart intervals. However, $time(|D_t|)$ is the time complexity to process $|D_t|$ incremental updates to me. Would'n you need to sum up the time complexities for each restart interval / lifetime of an incremental mechanism separately instead of bounding them by $|D_t|$, which may be even 0?

The notation $\\alpha^{(k)}$, in particular the definition of $(k)$, is a bit informal. The extensive list of arguments is very helpful, but the meaning of the stream-related exponent $k$ for a static algorithm is a bit ambiguous or vague.

The definition of union-preserving might deserve to be more prominent than a footnote.

**Ethical Concerns:**

["NO or VERY MINOR ethics concerns only"]

**Final Justification:**

The authors provided answers to all my (minor) questions. I still recommend acceptance.

**Limitations:**

Yes, all limitations addressed.

**Paper Formatting Concerns:**

-

**Quality:**

3

**Strengths And Weaknesses:**

The algorithm is based on the established, high-level concept of maintaining the dynamic updates and privacy budget of a DP datastructure in a tree of logarithmic depth, but generalizes it by keeping track of insertion/deletion pairs represented as intervals. The authors propose an online version of an interval tree, where the index of a node in an in-order traversal is $t$. An insertion at time $t$ is tracked on the path from the root to the node $v_t$, and it is removed from it at the time of deletion. At each node, the algorithm uses three black-box mechanisms: a static DP mechanism for $f$ to provide an answer for the dataset at time $t' \\leq t$, an insertion-only DP mechanism to track deletions since $t'$, and a insertion-only DP mechanism to count deletions. The latter mechanism is required because when a node's privacy budget is exceeded, it needs to restart its mechanism (and sets $t' = t$).

The main technical contribution and novelty of this paper is a black-box construction for dynamic data streams and union-preserving queries. Except for a few special cases like counting (where $f \\equiv 1$), it reduces the best known multiplicative error bound for linear queries in the number of updates $t$ from linear to $log^2$. Arguably, counting is an important special case, and the improvement for this case is $\\log^2.5$ to $\\log^2$. The paper is well written except for a few minor improvements to notations that could be made, and appears to be sound on the level of arguments (without checking numbers).

---

> ### Author Rebuttal · Authors · 2025-07-26
>
> > Q1. The running time as stated is a bit confusing to me...
>
> Thank you for pointing this out. This is a typo: We meant to write $O(\log^2 t) \cdot time(N_t)$ for the running time, where $N_t=|D_t^+|+|D_t^-|$ is the number of updates (rather than the data size). The two $\log$ factors account for the fact that each node can be restarted $(\log t)$ times, and the tree has height $(\log t)$. Within each round, the number of items is bounded by the number of updates.
>
> > Q2. The notation $\alpha^{(k)}$, in particular the definition of $(k)$, is a bit informal. The extensive list of arguments is very helpful, but the meaning of the stream-related exponent $k$ for a static algorithm is a bit ambiguous or vague.
>
> We would like to clarify that $\alpha^{(k)}$ is not necessarily defined with respect to a stream. It just refers to the total error when summing up $k$ DP estimators.  For example, when answering a range query consisting of $k$ counts from a standard Laplace histogram, the error will also be $\alpha_{\mathrm{Lap}}^{(k)}=\tilde{O}(\sqrt{k})$. We just use this notation (instead of $k\cdot \alpha$, which is not tight) to express our error in the streaming setting.
>
> > Q3. The definition of union-preserving might deserve to be more prominent than a footnote.
>
> We will move it to the main text.

---

> > ### Comment · Reviewer_QjJv · 2025-08-02
> >
> > Thank you for the clarifications! Regarding Q2, the definition makes perfect sense like this, I read it as stream-specific notation somehow.

---

### Official Review · Reviewer_Px2a · 2025-06-30

**Clarity:** 3
**Significance:** 4
**Originality:** 3
**Rating:** 5
**Confidence:** 3

**Summary:**

This paper introduces a new black-box mechanism that integrates any given algorithm for the private linear query problem with the binary tree mechanism to construct an algorithm for private query release under continual observation. This method achieves lower additive error for the streaming turnstile model than that indicated by prior work. For context, the private linear query problem simply asks the data curator to return the result of an arbitrary set of linear (counting) queries under the constraints of $\varepsilon$-differential privacy - prior work shows that this is possible with additive error $\sqrt{n}/\varepsilon$, where $n$ is the size of the private data set. The continual observation setting requires the data curator to accept modifications to the data set (insertions, deletions, or no change) in a stream, and respond to the set of queries after every update. A general-purpose technique in the privacy literature to deal with queries with an additive structure (union-preserving, as mentioned by the authors) is to use the binary mechanism of Chan et. al., that essentially allows one to use batch algorithms for the query release problem and suffer additive error that is inflated by a factor that is only polylogarithmic in the length of the stream, despite making a release after each new stream element (naive privacy accounting would lead to a $\sqrt{T}$ additive error when the stream is of length $T$ instead which is much worse).

It is observed by the authors that for the turnstile setting, this general purpose method suffers the drawback that the net error is not $\sqrt{n}\log^c T$ for some $c$ but instead on the order of $(\sqrt{n_+} + \sqrt{n_{-}}) \log^c T$, where $n_+$ s the number of insertions and $n_-$ is the number of deletions. For long streams, clearly this is much (potentially arbitrarily) worse than the former. The main contribution of this paper is a new algorithm that achieves additive error with only a $\sqrt{n}$ term.

Methods: The algorithm suggested is essentially a modification of the binary tree mechanism (maintaining a copy of the batch algorithm at each internal node of a binary tree over the stream), where one backtracks to tag nodes with 'liveness' interval endpoints when deletions are made. Just as in the binary tree mechanism, multiple copies of an element are maintained at each level, so are they here, but instead of each element only being added to the nodes corresponding to the dyadic intervals in which it lies, it is added to all nodes whose interval happens to lie within the liveness interval of that element. This leads to the possibility of double counting when a query is made, to deal with this one considers only the responses from the batch algorithm at `left' nodes.

To achieve the main improvement, the algorithm refreshes the privacy budget of each internal node when half the elements it carried previously are deleted, which ensures that the additive error remains on the order of $\sqrt{n}$. The restart points themselves can be privatized with the Sparse Vector Technique (SVT). Refreshing the budget in this manner is not too wasteful because this can happen only logarithmically many times for any node over the course of the stream.

**Questions:**

1. What happens if an element is deleted more often than it is inserted? It looks like it doesn't make much of a difference since the algorithm would just check the internal nodes and find nothing to add a deletion tag to, but this might be good to discuss explicitly.

**Ethical Concerns:**

["NO or VERY MINOR ethics concerns only"]

**Final Justification:**

I didn't have any major concerns for this paper, and there were no significant weaknesses to address, so I keep my score as is (a 5).

**Limitations:**

yes

**Paper Formatting Concerns:**

No concerns.

**Quality:**

3

**Strengths And Weaknesses:**

Strengths:
1. The problem is very general, of interest to the privacy community and impactful.
2. The algorithm is simple and straightforward to understand, intuition is given well.

Weaknesses:
1. I think the proofs need to be written more rigorously, possibly in the appendix - at a high-level this approach makes sense, but for instance not having a formal definition of a left node and a right node made the reading a bit harder.
2. Although intuition is given well, the writing could be polished a lot more - this is a very minor and subjective issue but subheadings like 'Putting it Together' are maybe a bit informal and not very helpful to a reader trying to find the right subsection quickly, for instance.

---

> ### Author Rebuttal · Authors · 2025-07-26
>
> > W1. I think the proofs need to be written more rigorously, possibly in the appendix - at a high-level this approach makes sense, but for instance not having a formal definition of a left node and a right node made the reading a bit harder.
>
> We will include formal definitions for these terms and revise the proofs.
>
> > W2. Although intuition is given well, the writing could be polished a lot more - this is a very minor and subjective issue but subheadings like 'Putting it Together' are maybe a bit informal and not very helpful to a reader trying to find the right subsection quickly, for instance.
>
>  We will change the subheading to 'Full Algorithm' and polish the writing.
>
> > Q1. What happens if an element is deleted more often than it is inserted? It looks like it doesn't make much of a difference since the algorithm would just check the internal nodes and find nothing to add a deletion tag to, but this might be good to discuss explicitly.
>
> Similar to our definition in line 14, all existing work on the turnstile model under DP (e.g. [18]) has also adopted the **strict** trunstile model. Under this model, an item to be deleted must exist in the dataset. A more relaxed model, which allows negative counts, has been studied in the non-private setting, but it has not been studied under DP yet.

---

### Decision · Program_Chairs · 2025-09-17

**Decision:**

Accept (spotlight)

**Comment:**

This paper gives a black-box method that can be used to go from a private linear query algorithm to a private query release algorithm that works in dynamic streams where elements may be inserted or deleted. The construction presented improves better parameters than existing algorithms and works on fully dynamic streams in contrast with insertion-only streams. The paper achieves nearly optimal dependence on the length of the stream (up to polylog factors).
The reviews do not identify any significant weakness except recognizing that these techniques are unlikely to work beyond linear (or more specifically, a union-preserving family) or queries.
I think this paper is quite impressive because of the generality and (near) optimality of the result, resolving questions that several papers had looked at. In addition, some reviewers thought that the paper was well written. Because of these reasons, I would like to recommend this paper for a spotlight.
The discussion was collegial. The issues identified by the reviewers were very minor, and addressed well by the authors.